# Empowering Future Physicians and Communities on Chicago’s South Side through a 3-Arm Culinary Medicine Program

**DOI:** 10.3390/nu15194212

**Published:** 2023-09-29

**Authors:** Geeta Maker-Clark, Ashley McHugh, Hannah Shireman, Valeria Hernandez, Megha Prasad, Tiffany Xie, Arianna Parkhideh, Carlin Lockwood, Sonia Oyola

**Affiliations:** 1Department of Family Medicine, NorthShore University HealthSystem, Evanston, IL 60201, USA; 2Department of Family Medicine, University of Chicago, Chicago, IL 60637, USA; 3Pritzker School of Medicine, University of Chicago, Chicago, IL 60637, USA

**Keywords:** community health, culinary medicine, medical education, nutrition education

## Abstract

The purpose of this pilot evaluation was to assess the impact of a university culinary medicine program on participating medical students and community members, which included individuals managing chronic illness and public middle school students. A total of 59 program participants enrolled in the study. Data were obtained using pre- and post-course surveys and qualitative interviews from September 2021–July 2023. Results show increased confidence in medical students’ ability to provide nutrition counseling, with a high significance in their ability to provide counseling regarding chronic conditions. Participants managing chronic conditions demonstrated significant increases in self-reported confidence in their understanding of overall chronic disease management and care and in their kitchen skills, with participants who attended five or more classes having significantly higher means. Qualitative feedback from middle school students highlights their knowledge and willingness to try new foods after engaging with the curriculum. Findings add to the growing literature on culinary medicine and provide insight into the effectiveness of culinary medicine programming to increase knowledge and promote positive changes among future healthcare professionals and community members. However, more extensive research across a longer time span is needed to confirm the potential for sustained change.

## 1. Introduction

In recent years, growing evidence has demonstrated the connection between nutrient-dense food and better health outcomes and, furthermore, nutrient-poor diets have emerged as a leading risk factor for all-cause mortality [1,2,3]. The field of culinary medicine has emerged as an innovative and significant approach to promote better health outcomes by integrating evidence-based nutrition knowledge with practical culinary skills, as well as training healthcare professionals in dietary counseling. By leveraging the power of food and cooking as tools for prevention and treatment, culinary medicine has gained traction in various healthcare settings, leading to improved patient outcomes and enhanced quality of life [4,5].

The literature has shown a powerful interest in the use of culinary medicine as an educational tool. A scoping review by Tan et al. found 24 studies between 2020 and 2022 that involved a hands-on culinary medicine component to nutrition, speaking to the growth of research and implementation in this innovative field of medicine within a short period of time [6]. In medical education, there is an ever-growing number of offerings to train future physicians in this blend of culinary art and nutritional science, including ours at the University of Chicago [7]. While traditional nutrition curricula are based on lecture and classroom teaching, culinary medicine programs amplify the role of food as medicine for improved health through experiential, hands-on learning in a teaching kitchen setting. Interdisciplinary teams of physicians, chefs, and nutrition experts unite their skills to make evidence-based recipes that promote healthful, flavorful eating.

While culinary medicine holds promise for improved nutrition education for health professionals, its potential impact on “at-risk” populations is particularly significant. At-risk populations are defined as those who face a higher risk of poor health outcomes due to a combination of socioeconomic, environmental, and systemic factors [8]. These populations may include low-income individuals, racial and ethnic minorities, immigrants, refugees, the elderly, and individuals experiencing homelessness or food insecurity. The South Side of Chicago has significant population-level disparities within social care and healthcare systems. One noted example is the 30-year gap in life expectancy between people living in poor, predominantly African American neighborhoods on the South Side, as compared to those in the more affluent, predominantly white neighborhoods only nine miles away. This is considered the largest life expectancy gap in the US, and significantly contributes to fewer economic opportunities and higher rates of obesity, diabetes, hypertension, heart disease, and stroke in South Side neighborhoods [9,10].

Health disparities, characterized by marked differences in health outcomes and access to healthcare services, persist among at-risk populations [11]. These disparities are often rooted in complex social determinants of health, including limited access to nutritious food, cultural barriers, educational disparities, and economic challenges. The integration of culinary medicine into the care and support of historically marginalized populations offers a needed strategy to address these disparities by fostering a holistic approach to health that not only focuses on the treatment of diseases but also prioritizes prevention and health promotion through nutrition education tailored to the community [12]. This pilot study aims to evaluate how a culinary medicine program can be applied in three diverse settings to amplify nutrition education leading to behavioral change. Our program initially commenced in 2015 as a medical school elective and has grown to include a cooking and nutrition class for patients and community members with chronic disease and a course for middle schoolers in a Chicago public school. By describing our three-armed model and presenting the results of this evaluation, we highlight the potential of culinary education programs to address the unique needs of various populations within a community.

We shed light on the benefits associated with implementing culinary medicine interventions and share unique considerations to address when implementing culinary medicine initiatives with partnering communities. These include cultural humility and decolonization of nutrition education, affordability, accessibility of fresh and nutritious ingredients, food preferences and restrictions, and the creation of sustainable community partnerships. Our work provides evidence of impact for historically marginalized populations and highlights strategies we have used to create programming specific to communities on Chicago’s South Side.

## 2. Materials and Methods

### 2.1. Study Design and Participants

This is a pilot evaluation of the University of Chicago Culinary Medicine program from September 2021–July 2023. Participants of this cohort study include fourth-year medical students who participated in the Medical Student Elective, middle-school-aged learners who participated in the Food Is Power program through a local public school, and adult community members managing chronic disease who participated in Community Cooking & Nutrition classes. Study participants are therefore representative of the program participants. All program participants were given the opportunity to participate in the research prior to program participation and informed consent was obtained from all subjects involved in the study. Research participation was not a requirement of program participation. This study was reviewed and approved by the University of Chicago Institutional Review Board (IRB), NorthShore University IRB, and Chicago Public Schools Research Review Board.

### 2.2. Culinary Medicine Program and Curriculum

The University of Chicago Culinary Medicine program, created by two practicing integrative medicine physicians (SO and GMC), utilizes a three-armed approach to educate future physicians and nearby communities about the importance of nutrition, its health impacts, and how to utilize nutrition literacy for improved health.

#### 2.2.1. Medical Student Elective

The Medical Student Elective for fourth-year medical students consists of 8 sessions co-taught by a faculty physician (SO) and a professional chef, along with guest faculty who join to co-lead specific content areas. Each session has corresponding pre-class readings, a discussion, and a hands-on cooking lesson in a teaching kitchen. This course is patient- and cooking-centered and teaches interactive counseling and chef-taught kitchen skills. Topics include: (1) Intro to Culinary Medicine, (2) Sports Nutrition, (3) Fats, (4) Sodium, Potassium, Hypertension, (5) Nutrition in Pregnancy, (6) IBS, IBD and GERD, (7) Proteins, Amino Acids, Vegetarian Diets and Eating Disorders, and (8) Cancer Nutrition. These modules are licensed from Health meets Food: The Culinary Medicine Curriculum. The Medical Student Elective adds clinical skills through education in nutrition compared to the traditional medical school curriculum, which generally only includes didactic lectures in the pre-clinical years through biochemistry coursework.

#### 2.2.2. Food Is Power

Food Is Power is a program designed for middle-school-aged learners and is taught by a faculty physician (GMC), chef, and medical students who demonstrate interest in culinary medicine. This program teaches a decolonized, bespoke curriculum written for the South Side community. The curriculum, created with input from community leaders, local urban agriculture organizations, and school faculty, addresses food justice and the roots and routes of food and provides a robust nutrition curriculum to students residing in an area that has long been subjected to disinvestment, redlining, and systemic racism. It is designed to be flexible and adaptable with customized recipes, cooking demonstrations in every class, and colorful educational materials that align with the specific cultural and dietary needs of the community. This ensures that participants can relate to the content and feel empowered to make healthier food choices while honoring their cultural heritage. The program rises to the unique challenges of youth growing up with food insecurity, specific cultural food practices, and the need to create nutrition literacy, as well as providing an extra meal on class day. All participants are middle-school-aged students at a Chicago public school.

#### 2.2.3. Community Cooking & Nutrition

The Community Cooking & Nutrition classes for adult community members managing chronic disease are designed to address the chronic disease epidemic through a nutrition education lens. Taught by a faculty physician (author SO), chef, and medical students interested in culinary medicine, participants learn how to mitigate food allergies, work with diabetic diets, create culturally sensitive meals, and prepare healthy food on a budget that is local, nutritious, and delicious. All participants are residents of the South Side of Chicago or patients at the University of Chicago.

### 2.3. Material Design

#### 2.3.1. Medical Student Elective

A 55-item survey was administered to medical students both prior to the first class and upon completion of the course. Questions were divided into 4 sections: perceived importance of incorporating nutrition counseling into clinical care, personal dietary habits, perceived competence in ability to provide nutrition counseling, and demographic information. To assess the students’ perceived importance of incorporating nutrition counseling into clinical care, students were given statements about provider involvement in nutrition counseling and asked to rank the extent to which they agreed using a 5-point Likert scale. To assess personal dietary habits, students were given a list of various foods and asked to indicate how frequently they consumed each food on average throughout the past six months, using the following scale: 1 = “Never”, 2 = “One to two times per week”, 3 = “Three to six times per week”, 4 = “Seven times per week (daily)”, 5 = “Two or more times daily”. To assess perceived competence in ability to provide nutrition counseling, students were given the question stem “For me, educating patients independently of support from other medical professional on the following topics, I feel…” and asked to use a 5-point Likert scale to rank their confidence in their ability to counsel patients on specific diets, management strategies for chronic diseases, components of food, and basic nutrition literacy skills. To assess demographic information, students were asked a variety of multiple choice and extended response questions regarding their identities, current level of medical training, and personal experiences related to nutrition counseling.

#### 2.3.2. Food Is Power

Middle-school participants were given printed surveys, in class, prior to participation in the course and following the final session. In the 2021–2022 academic year, the pre- and post-surveys contained 41 questions, which were condensed into an 18-item pre- and post-survey in the 2022–2023 academic year due to the length of time to completion. The surveys assessed students’ interest and knowledge in nutrition and cooking by indicating the extent to which they agreed with statements about nutrition and cooking using a 5-point Likert scale. Food is Power-specific goals, such as empowering students to make healthy food choices and increase in nutrition literacy, were assessed via “Yes/No” style questions about personal nutrition skills and knowledge and frequency questions to assess their nutrition habits and attitudes. The pre-survey contained demographic information through multiple choice questions with fill-in-the-blank options, if needed. The demographic data were removed in the post-survey and replaced with extended response questions assessing impact and the students’ favorite parts of the class.

#### 2.3.3. Community Cooking & Nutrition

A 9-item written survey was administered to participants prior to their first community cooking class and upon completion of the final class that they attended. The surveys assessed participants’ confidence in their ability to make healthy food choices, priorities when choosing what to eat, typical dietary habits, and goals for the cooking course. Confidence in choosing and preparing healthy foods was measured by providing statements about nutrition management and having participants rank the extent to which they agreed using a 5-point Likert scale. Participants’ priorities when making food-related decisions were assessed by providing a list of options with the instruction to select all that apply, along with the option to specify any priorities that the list did not cover. Typical dietary habits were elicited via short free response questions. Participants’ goals for the program were elucidated via a select-all-that-apply list of options and a free response question asking about specific nutrition concerns. The pre-survey included questions on demographic information whereas the post-survey included perceived impact of the course. Impact was evaluated through multiple choice and free response questions asking what participants gained from the program, how they had been affected by the program, and what aspects of the program needed improvement. The surveys were revised between the 2022 and 2023 cohort. The questions remained the same, but certain multiple choice questions had additional answer choices in the 2023 version.

The 2023 participants were also given the opportunity to participate in a qualitative interview with a member of the research team (MP). Interviews lasted approximately 13 min and were conducted via phone or Zoom according to participant preference. Interviews included questions surrounding knowledge gained from classes, priorities in decision making about what to eat, managing health condition(s), and access to ingredients and supplies used during classes.

### 2.4. Analysis

Survey data were collected directly from REDCap in the Medical Student Elective. In Food Is Power and the Community Cooking & Nutrition classes, they were first collected via a written survey and then imported to REDCap before being described and analyzed. For each arm, paired data were pooled between the 2021–2022 and 2022–2023 collection years. Quantitative analyses were conducted using Stata Release 17 (StataCorp. 2021. Stata Statistical Software: Release 17. College Station, TX, USA: StataCorp LLC). Given small sample sizes, we calculated exact *p*-values within these groups. Qualitative data were analyzed using Dedoose.

#### 2.4.1. Medical Student Elective

Likert scale questions were analyzed using the Wilcoxon signed-ranks test.

#### 2.4.2. Food Is Power

The 2021–2022 survey had 41 items whereas the 2022–2023 survey was abbreviated to 18 items. For this analysis, only data from common questions were pooled. Three questions appeared as 5-point Likert-style questions on the 2022–2023 survey that were originally asked as “Yes/No” questions in the 2021–2022 survey. For these questions, “Strongly Agree” and “Agree” were coded as “Yes”, “Strongly Disagree” and “Disagree” were coded as “No”, and those who selected “Neutral” were excluded from analysis since we could not determine which side of the dichotomy respondents were leaning towards. Eight of the common questions were originally presented as “Yes/No” questions but were later changed to include a third option (“Not Sure”). Because this third option did not exist in the 2021–2022 survey, we excluded respondents who answered “Not Sure” from analysis to make the comparison across surveys more equal. In the 2021–2022 survey, there was one “Always/Sometimes/Never” question that was changed into a 5-point Likert scale in the 2022–2023 survey. For this question, we attempted to group the 5-point Likert scale in a comparable way—we did not change responses for “Always” and “Never”, but we did group “Rarely”, “Sometimes”, and “Usually” as “Sometimes”. Two questions from the 2021–2022 survey were 4-point Likert scales that were expanded to 5-point Likert scales in the 2022–2023 survey. To make responses comparable, we excluded “Neutral” from analysis and changed the direction of the 2022–2023 scale. Resulting “Yes/No” questions were analyzed using the McNemar’s chi-square test without continuity correction and Likert-style questions were analyzed using the Wilcoxon signed-ranks test.

#### 2.4.3. Community Cooking & Nutrition

Likert questions were analyzed using the Wilcoxon signed-ranks test. Using post-survey responses to those Likert-type questions, we performed Welch *t*-tests comparing groups created using participants’ self-reported attendance. One group was made up of participants who had attended fewer than or equal to 4 classes, and the other group were those who attended 5 or more classes. To compare what participants hoped to gain to what they did gain from the classes, we placed responses in two-way tables.

Interviews were recorded, de-identified, and transcribed before being uploaded to Dedoose for coding. A codebook was developed to analyze key elements of the interviews. Each transcript was coded separately by two members of the research team (MP and HS) and coding discrepancies were resolved through iterative discussions between the two coders.

## 3. Results

Research study participants (N = 49) were representative of culinary medicine program participants overall. Fourteen Medical Student Elective participants chose to participate in the pre/post-surveys, along with eleven Food Is Power program participants and twenty-four Community Cooking & Nutrition class participants. Additionally, 71.4% of 2023 Community Cooking & Nutrition class participants also opted to participate in qualitative interviews. Demographics of program participants are reported in Table 1.

### 3.1. Medical Student Elective

Seventy-nine percent of medical student participants identified as female with a mean age of 27.8 years. Half of participants identified as non-Hispanic White, 29% as non-Hispanic Asian, 14% as non-Hispanic Black, and 7% as other. Students came to the course from a broad range of intended specialties, including family medicine, internal medicine, pediatrics, gastroenterology, anesthesia, emergency, pathology, and psychiatry. Before completing the elective, 86% of participants had no nutrition training before medical school.

Students began the elective with positive beliefs and attitudes about nutrition education in medicine, with most (13/14) indicating that they agree or strongly agree that specific advice about how to make dietary changes could help patients improve eating habits. Participants showed a positive shift in the belief that nutritional counseling should be included in any routine appointment, just like diagnosis and treatment (*p* = 0.031).

Generally, students reported increased confidence in nutrition counseling after participating in the course. Figure 1 displays that students’ confidence in their ability to provide nutrition counseling regarding several chronic conditions increased significantly, including type 2 diabetes (*p* < 0.001), celiac disease (*p* = 0.001), osteoporosis (*p* < 0.001), eating disorders (*p* = 0.027), and food allergies (*p* = 0.001). Students also reported significantly increased confidence in terms of their ability to educate patients on specific dietary patterns and their health effects, like the Mediterranean diet (*p* = 0.002), DASH diet (*p* < 0.001), vegetarian diet (*p* < 0.001), very-low-fat diet (*p* < 0.001), and high-protein/high-fat diet (*p* = 0.002) (see Figure 2).

Moreover, there were significant increases in students’ confidence in their ability to counsel patients on general aspects of a typical diet, like examples of MyPlate serving sizes (*p* = 0.003), the health effects of moderate alcohol consumption (*p* < 0.001), the role of dietary cholesterol and saturated fat (*p* < 0.001), the role of omega-3 and -6 fatty acids in heart health (*p* < 0.001), the role of the glycemic index (*p* < 0.001), the role of fiber (*p* < 0.001), the role of hydration and fluid needs based on activity and age (*p* < 0.001), how to identify antioxidant-rich grocery produce (*p* = 0.001), and ability to read and analyze food labels (*p* = 0.016). Students’ own eating behaviors did not change significantly between the pre- and post-tests.

### 3.2. Food Is Power

Six students identified as female, two students identified as male, one identified as non-binary, and two chose “prefer not to answer” or “other”. The majority of participants (10/11) identified as Black/African American, and the mean age was 12.7 years.

The adjustment to pre- and post-surveys between data collection periods and the study’s small sample size made it challenging to calculate statistical significance for this subsection of research participants. However, the qualitative feedback shared by students on post-surveys speaks to the impact of the class. Qualitative results center on two main themes: understanding of the “food is power” concept and participant’s knowledge and willingness to try new foods. When asked about the meaning of food is power to them, students highlighted that food can provide energy and that food choices are important. Participants also reported learning to cook and try new foods, which was an aspect of class that they found particularly enjoyable (see Table 2).

### 3.3. Community Cooking & Nutrition Classes

Twenty-four participants in the Community Cooking & Nutrition classes elected to participate in the pre-/post-surveys from 2022 (n = 5) and 2023 (n = 19). Of these participants, 87.5% identified as female and 12.5% as male; 83.3% identified as non-Hispanic Black, 4.2% as both non-Hispanic Black and Native American, 8.4% as non-Hispanic White, and 4.2% as “other”. Survey participants had a mean age of 67.1 years. Fifteen of the 2023 participants also chose to participate in a qualitative interview. Of these participants, 86.7% identified as non-Hispanic Black and 13.3% identified as non-Hispanic White; 93.3% of interview participants were female with a mean age of 66.4 years.

In comparing survey responses from before their first class to after their last class, participants demonstrated significant increases in self-reported confidence in their understanding of overall chronic disease management and care (*p* = 0.005) and in their kitchen skills (*p* = 0.02). Participants also seemed better equipped to make more nutritious changes in their diets as their self-reported confidence in knowing what foods to eat to manage their own health condition(s) increased (*p* = 0.007) and it became easier for them to make healthy food choices (*p* = 0.001; see Figure 3a).

Participants who attended five or more classes had significantly higher means in post-survey Likert scale responses relative to participants who attended four or fewer classes (see Figure 3b). Participants who attended five or more classes were more confident in their kitchen skills (*p* < 0.001) and in their understanding of overall chronic disease management and care (*p* = 0.012). These participants also reported significantly increased knowledge in managing their health condition(s) with food (*p* = 0.013) and greater ease in making healthy food choices (*p* = 0.012). The more frequent participants also had a significantly higher agreement that their nutrition concerns were addressed in the class (*p* = 0.029).

Participants were asked what they hoped to gain from the classes in the pre-survey and what they actually gained in the post-surveys. Of the 21 participants who hoped to gain new recipes they could use at home, 21 did. Twenty-two participants reported gaining kitchen skills and techniques, nine of which had not initially hoped to gain these skills from the class. Thirteen of the seventeen participants who originally wanted to gain confidence in choosing foods that were right for them reported gaining that confidence. Sixteen of the nineteen participants who hoped to gain tips on managing challenges to healthy eating did so. While the remaining three did not indicate this gain, four additional participants who did not anticipate gaining tips did (see Table 3). The 2023 surveys added three new, potential gains, so only 19 participants were able to see and provide answers for those additions. Of the 15 who wanted to gain a better understanding of how to manage their own health condition(s) using food, 11 did. Twelve participants reported gaining a better understanding of general management of their health condition(s) with one who had not hoped to gain that from the class. Seventeen participants reported gaining a sense of community and belonging—eight of which had not initially set out to do so.

The 19 participants in the 2023 class were asked about access to resources used in making class recipes. Eighteen participants agreed or strongly agreed that they could easily get to a grocery store that sells the ingredients used in class recipes. Seventeen participants agreed or strongly agreed that they have time to make the class recipes at home and that the class recipes would be easy for them to make at home. Sixteen participants agreed or strongly agreed that the class recipes are affordable to make at home and that they already owned or could easily purchase the kitchen tools used in class recipes. Out of all 24 participants, 23 agreed or strongly agreed that they would share what they learned in the classes with their family and community.

When asked in qualitative interviews why they decided to join the Community Cooking & Nutrition classes, participants shared that health concerns and a desire to learn about how to make healthier food choices were their primary reasons for taking the classes. Interviewees largely felt these needs were addressed through the provided lessons. Roughly 3/10 participants discussed learning about the glycemic index, or how much specific foods increase their blood sugar, without being primed or directly asked. This is key knowledge in management of chronic conditions such as diabetes and even pre-diabetes, which are conditions that 60% of participants reported that they or a family member have in their interview. The interview did not ask specifically about what health condition(s) participants have, and offering of this information was voluntary. Thus, this number does not include other individuals with diabetes and pre-diabetes in the class.

The class helped participants learn about new foods and healthier alternatives to what they currently make at home. This was particularly important, as many participants elaborated that learning about new food options that are easy to prepare and incorporate into meals that they already know how to make has motivated them to choose healthier options more frequently. Interviewees reported these behavioral changes as ones they have already made or planned to make related to their diet. As one participant said, “You know, I will choose the options that are healthier for me because I never knew they could taste so good. And now that I know it makes it easier to eat the healthy foods instead of the unhealthy foods. I want to eat more of the vegetables now”.

In addition to the nutrition knowledge gained, participants highlighted the feelings they had after taking the classes as being encouraged and excited to use their new knowledge to help manage their conditions moving forward. The sense of empowerment gained from the classes was bolstered by the sense of community created amongst class participants. As summarized by one participant, “Coming to this class and the camaraderie we had with meeting other people and everybody sharing their stories. And we all had, you know, we all had our stories and our struggles. Because truly, truly diabetes, it is-it is a struggle”.

## 4. Discussion

The findings of this study provide insights into the effectiveness of culinary medicine programs in promoting positive changes among future healthcare professionals and community members. The Medical Student Elective is designed to train future physicians to support the very communities participating in the patient- and middle-school-centered classes. Culinary medicine training for medical student participants at our institution increased self-reported confidence in discussing nutrition in patient care settings and heightened awareness of the impact of nutrition on health outcomes. These findings echo other studies that observed increases in dietary counseling confidence among medical students participating in nutrition education [13,14]. It is possible that this optional Medical School Elective attracted students with a higher level of knowledge and interest in nutrition. However, results of the evaluation show a significant increase in confidence in nutrition counseling, for general health and several commonly seen chronic conditions. Students also acknowledged the relevance of nutrition education in their future medical practice. This aligns with the resounding need for healthcare providers to be equipped with nutrition knowledge and communication skills to effectively address patients’ nutritional needs and the impact that culinary medicine courses can have on competency around nutrition counseling and knowledge [15,16,17]. While not formally measured, the medical students received the opportunity to gain culinary medicine teaching experience by supporting the patient-facing and middle school class lessons. This community integration enabled the medical students to have greater exposure to the curriculum and develop a deeper understanding of the community that we work in and the challenges that our community members face such as children experiencing food insecurity and homelessness, victims of gun violence, and adults with multiple chronic medical conditions. We feel that the medical students’ immersion into the community programs we serve was one of the most vital and significant aspects of this work.

The Food Is Power survey responses for the middle school class over the last year reflect that nutrition label reading literacy, understanding food as a medicine, understanding where food comes from, and exposure to the concept of rainbow colors of fresh food were principles that kids resonated with, though these findings were not significant. The recipes for this program are budget-friendly, made with easy to obtain ingredients in the community, do not require a stove, and are quick to prepare (under 20 min). Survey results were limited by small numbers, particularly while the COVID-19 pandemic affected school attendance and size of classes allowed, as well as the changes made to the survey between the first and second years of research. Testimonials reflected the practical importance of exposure to specific concepts and skill building in nutrition education, like reading labels and understanding of food as a source of empowerment and “like you can make powerful choices for your body”. The class is taught in the cafeteria classroom of a school without a hot food kitchen, and thus recipes do not require cooking on a stove. Considerations for teaching in a public school include requirements for parent consent to student surveys, ensuring surveys are short and efficient to allow for adequate class time, shorter attention spans in youth, having sufficient faculty on hand to supervise knife skills and hand hygiene, and having a space ideally within the school that can allow for cooking and teaching. Additionally, it is vital to build a strong and trusting relationship with school administration and teachers to be able to best serve the needs of students. Their input informs expectations of student energy, affect, and circumstances on teaching days which allows us to adjust accordingly to have the best class possible.

The Community Cooking & Nutrition class data showed that most participants were able to meet the goals that they stated in the pre-survey (i.e., gaining tips to manage challenges to healthy eating and getting new recipes to incorporate into their daily routines). Additionally, participants who attended five or more classes had significantly higher means in post-survey Likert scale responses relative to those who attended four or fewer classes. Participants who attended five or more classes felt greater ease in making healthy food choices, reporting increased knowledge in managing their health condition and more confidence in their kitchen skills. This finding may be due to several factors: as people attend more cooking classes, they might feel more comfortable experimenting with new recipes and cooking methods, have positive social interactions that foster a sense of community, and have additional motivation to practice cooking regularly and skill acquisition. Future culinary medicine courses for community members might consider being five or more sessions long to achieve the learning objectives. Additionally, recipes were made available to participants, a key element of aiding sustainable behavioral change, and an important element of culinary medicine course creation. The empowerment and community creation from these classes offered a social platform for participants to interact with others who share similar life challenges. This can lead to new friendships, a sense of belonging, and a stronger community connection, as well as an empowerment mindset to take better charge of chronic disease management. All participants stated that they would share what they learned with family and community, suggesting the “ripple effect” of knowledge shared in these classes resulting in a much larger impact on the community.

There are several strengths of this study. First, study participants reflect a range of populations that can benefit from the culinary medicine education model. Secondly, data represent two consecutive years of programming, during which course instructors and core curriculum remained unchanged. This continuity in programming allows for more robust data collection representing the program as a whole.

Several limitations should be noted. The self-reported nature of our data, particularly pertaining to dietary behaviors and attitudes, may introduce social desirability bias and the study’s relatively short follow-up period may not adequately capture long-term changes in behavior or attitude. As a result of these limitations, our results might not be generalizable and may be dependent on the participants’ demographic characteristics such as level of education, age, and urban geographic location. Our small sample of middle school students is of particular note, as we were not able to achieve significance for pre-/post-survey results. Surveys shared with middle school students were adjusted between data collection periods, which made it difficult to pool pre-/post-surveys between 2021 and 2022 as changes needed to make certain questions comparable meant the exclusion of some responses. Further study is required to develop standardized best practices and curricula particularly in these populations. Future research should consider employing objective measures of counseling competency and exploring the sustainability of behavior changes over an extended period. Extending the study period would better determine whether programming increases counseling competency beyond the medical school experience and into professional clinical practice.

## 5. Conclusions

In this study, we demonstrate that a culinary medicine curriculum can increase confidence in knowledge of general nutrition principles and counseling among fourth-year medical students. This has implications for medical education, patient care, and public health. Incorporating culinary medicine education into medical curricula could help foster a generation of healthcare professionals who are better equipped to promote healthy lifestyles and nutrition-related interventions among their patients. The positive changes, including an increased sense of confidence and willingness to try new foods, in community and middle school participants, show promise for culinary medicine to better equip community members to make informed, healthy food choices. We hope that these community- and student-based models can be valuable for other communities, especially those grappling with food insecurity, as adapting our program to other contexts with locally and regionally relevant foods and topics may achieve similar positive outcomes. Our results provide evidence for the importance and impact of nutritional literacy in communities. The potential impact of culinary medicine programs, particularly in a variety of teaching settings, is immense and includes improving health outcomes, enhancing food security, improving social connection and educational opportunities, reducing health disparities, strengthening patient–provider relationships, and community resilience.

This summary of our culinary medicine program may benefit institutions in the process of developing their own programming, particularly for the communities that they serve. By gaining a deeper understanding of the role of culinary medicine in addressing the health needs of vulnerable populations, healthcare providers, policymakers, and community leaders can work together to develop targeted interventions that can bridge the gap in health disparities and improve the well-being of those most in need.

## Figures and Tables

**Figure 1 nutrients-15-04212-f001:**
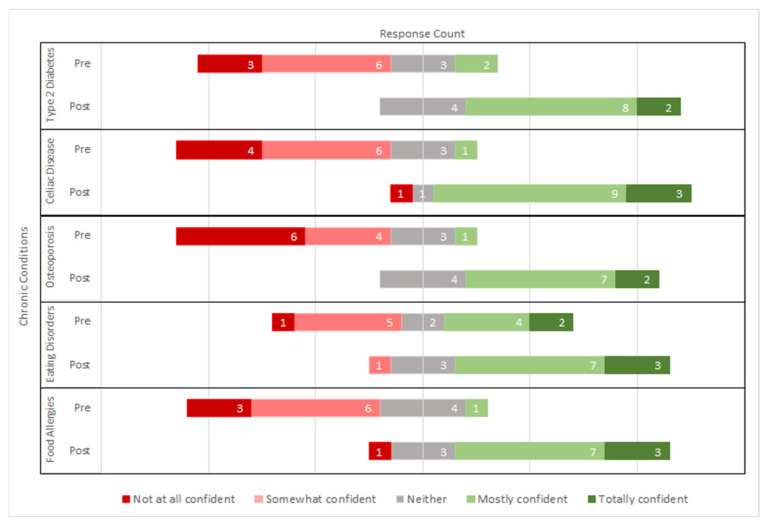
Change in medical student confidence in their ability to provide nutrition counseling on chronic conditions from pre- to post-assessment (N = 14).

**Figure 2 nutrients-15-04212-f002:**
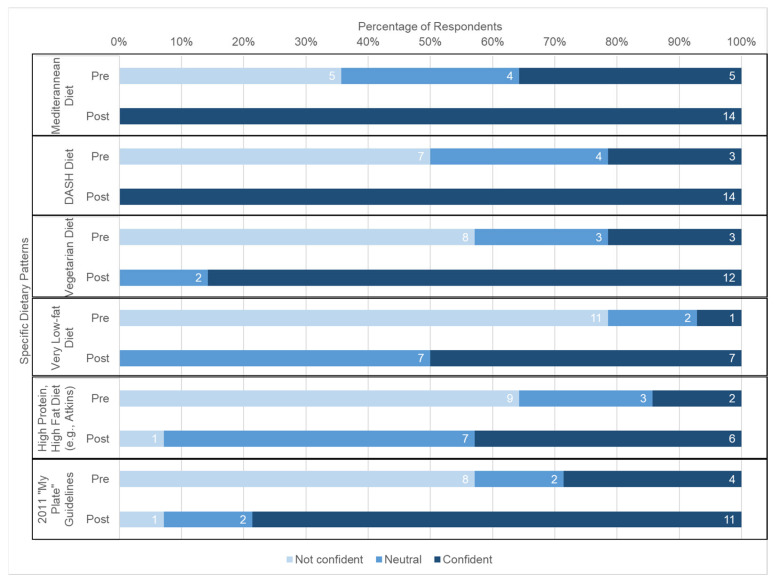
Change in medical student confidence in their ability to educate patients on specific dietary patterns, independent of support from other medical professionals, from pre- to post-assessment (N = 14).

**Figure 3 nutrients-15-04212-f003:**
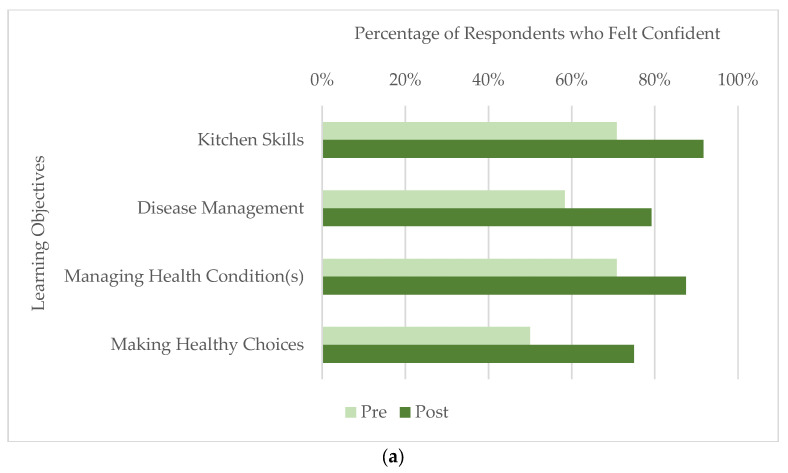
Self-reported change in confidence amongst Community Cooking & Nutrition participants (N = 24) from pre- to post-survey: (**a**) Percentage of respondents who agreed with being confident with each learning objective before and after class participation; (**b**) Difference in confidence, as measured by average Likert scale responses, between participants who attended 1–4 classes versus 5 or more.

**Table 1 nutrients-15-04212-t001:** Characteristics of Culinary Medicine program research participants (N = 49).

	Medical Student Elective (N = 14)	Community Cooking and Nutrition (N = 24)	Food Is Power (N = 11)
Mean age	27.8	67.1	12.7
Gender			
Female	11	21	6
Male	3	3	2
Non-binary	-	-	1
Prefer not to answer/Other	-	-	2
Race/Ethnicity			
Non-Hispanic Asian	4	-	-
Non-Hispanic Black	2	20	10
Non-Hispanic White	7	2	-
Non-Hispanic Black + Native American	-	1	-
Hispanic or Latino	-	-	1
Other	1	1	-

**Table 2 nutrients-15-04212-t002:** Qualitative feedback shared by Food Is Power research participants.

Understanding of “Food Is Power” Concept
“Food is Power to me means that food, health foods give you energy or power”. 2021–2022 student
“It means you can make powerful choices for your body”. 2022–2023 student
“It means that what you eat can affect you and is an important power of life”. 2022–2023 student
“I think it means that the food you eat is [your] power”. 2022–2023 student
“To me it means that food can help you in a lot of ways for your body”. 2021–2022 student
“That is was important for people to know what they eat so they can learn from what they eat”. 2021–2022
Knowledge and Willingness to Try New Foods
“What I have learned in Food is power is that even when you don’t like a particular food that you had in the past you can always try to improve it”. 2021–2022
“I learned to always try new foods. As many say ‘Don’t let anyone yuck your yum’”. 2021–2022 student
“…I get to learn how to cook different types of food”. 2022–2023 student
“My favorite parts was when we would learn the history of some foods and then be able to make something new out of it”. 2021–2022 student
“I would learn different tips about different ingredients and facts”. 2021–2022

**Table 3 nutrients-15-04212-t003:** Community Cooking & Nutrition participant responses to if they hoped to gain “Tips on managing challenges to healthy eating” versus if they actually gained them from the classes (N = 24).

Tips on Managing Challenges to Healthy Eating (N = 24)		Actual Gain	
		Yes	No
Anticipated Gain	Yes	16	3
	No	4	1

## Data Availability

The data presented in this study are available on request from the corresponding author (SO). Data are securely stored as per University of Chicago guidelines and not publicly available due to the confidential nature of the study.

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
