# Peer review of "Empowering Future Physicians and Communities on Chicago’s South Side through a 3-Arm Culinary Medicine Program"

_nutrients, 2023, doi:10.3390/nu15194212_

Round 1

Reviewer 1 Report

This study aims to highlight the impact of culinary medicine programs on healthcare professionals and community members, assessing changes in knowledge, confidence, and behavior related to nutrition and healthy eating.

The manuscript is well-written and broken in to an adequate number of sections and subsections. The researchers present valuable insights into the effectiveness of culinary medicine programs in promoting positive changes among healthcare professionals and community members.

The strengths of the study consist of different participant groups and assessment of behavioral change. However, the study is not without limitations, such as small sample sizes, self-reported data, a short follow-up period etc. Although they are mentioned, limitations should be taken into account more carefully. The authors should better state how these limitations affect their final results. For example these results might not apply for large scale studies and are heavily dependent on the participants demographic characteristics such as level of education, age, etc.

Author Response

Thank you for this thoughtful review of our manuscript. We appreciate your time and care with our work.

The request to take limitations into account more carefully and to better state how these limitations affect their final results was remedied by adding the following statements into the Discussion at line 491 : As a result of these limitations, our results might not be generalizable and may be dependent on the participants’ demographic characteristics such as level of education, age, and urban geographic location.

Thank you for helping us improve the manuscript in this way and ensure that the conclusions are supported by the results. 

Best,

Dr. Geeta Maker-Clark

Reviewer 2 Report

Dear authors,

The work is clear and well-written. I just have some minor comments.

How does the Medical Student Elective compare to the traditional nutrition curriculum? Please describe the latter so to provide some more insight.

Please check Figure 3b to be sure that all data is visible.

While I agree with you that it is important to acknowledge the impact on the participants, I feel that you should discuss more the possible impact of the knowledge and confidence gained with these programs on community. You only address the ripple effect for the Community Cooking & Nutrition.

Last, you obviously analyse the implications on the selected location, but how could this information help communities and policymakers in other regions, especially when talking about food deserts or locations without access to healthy options?

Best regards

Author Response

Thank you so much for your thorough and kind review of our manuscript. We appreciate your comments and made the following changes:

1. How does the Medical Student Elective compare to the traditional nutrition curriculum?  line 113-116: The Medical Student Elective adds clinical skills through education in nutrition compared to the traditional medical school curriculum, which generally only includes didactic lectures in the pre-clinical years through biochemistry coursework.

2. Please check Figure 3b to be sure that all data is visible.-Thank you, this was done

3. I feel that you should discuss more the possible impact of the knowledge and confidence gained with these programs on community and how could this information help communities and policymakers in other regions, especially when talking about food deserts or locations without access to healthy options?

Addressed this lines 513-521 We hope that these community and student-based models can be valuable for other communities, especially those grappling with food insecurity, as adapting our program to other contexts with locally and regionally relevant foods and topics may achieve similar positive outcomes. Our results provide evidence for the importance and impact of nutritional literacy in communities. The potential impact of culinary medicine programs, particularly in a variety of teaching settings, is immense and include improving health outcomes, enhancing food security, improving social connection and educational opportunities, reducing health disparities, strengthening patient-provider relationships, and community resilience.

Thank you again for your time and thought with our work. We hope these additions reflect the changes you hoped to see.

Dr. Geeta Maker-Clark